# A bioinspired hydrogen bond-triggered ultrasensitive ionic mechanoreceptor skin

Vipin Amoli[1,2], Joo Sung Kim[1,2], Eunsong Jee[1,2], Yoon Sun Chung[1], So Young Kim[1], Jehyoung Koo[1], Hanbin Choi[1], Yunah Kim[1] & Do Hwan Kim [1]

Biological cellular structures have inspired many scientific disciplines to design synthetic structures that can mimic their functions. Here, we closely emulate biological cellular structures in a rationally designed synthetic multicellular hybrid ion pump, composed of hydrogen-bonded $[EMIM^+][TFSI^-]$ ion pairs on the surface of silica microstructures (artificial mechanoreceptor cells) embedded into thermoplastic polyurethane elastomeric matrix (artificial extracellular matrix), to fabricate ionic mechanoreceptor skins. Ionic mechanoreceptors engage in hydrogen bond-triggered reversible pumping of ions under external stimulus. Our ionic mechanoreceptor skin is ultrasensitive ($48.1$–$5.77$ $kPa^{-1}$) over a wide spectrum of pressures ($0$–$135$ kPa) at an ultra-low voltage ($1$ mV) and demonstrates the ability to surpass pressure-sensing capabilities of various natural skin mechanoreceptors (i.e., Merkel cells, Meissner's corpuscles, Pacinian corpuscles). We demonstrate a wearable drone microcontroller by integrating our ionic skin sensor array and flexible printed circuit board, which can control directions and speed simultaneously and selectively in aerial drone flight.

[1] Department of Chemical Engineering, Hanyang University, Seoul 04763, Republic of Korea. [2] These authors contributed equally: Vipin Amoli, Joo Sung Kim, Eunsong Jee. Correspondence and requests for materials should be addressed to D.H.K. (email: dhkim76@hanyang.ac.kr)

An ionic skin composed of deformable ionic materials represents a new class of deformable sensory platforms to emulate the tactile sensing features of human skin for potential applications in artificial skin technology[1,2]. Many ionic conductors such as ionic liquids (ILs), ionogels, and hydrogels have been used to implement the ionic skin with human skin-like perceptive characteristics[3–6]. Therefore, the ionic skin can effectively sense pressure, strain, shear, torsion, and other external stimuli, but struggles to maintain high sensitivity over a wide spectrum of pressures in task-specific applications such as robotics and prosthetics[1–7]. Recently, biomimetics has emerged as a burgeoning area in artificial skin technology that has led innovations in material designing and device structure manipulation with the aim to imitate tactile sensing features of human skin intelligently[8–12]. Biological cellular structures are the source of inspiration because of their intriguing structural and functional properties[11,12].

In general, humans are multicellular organisms consisting of various types of living cells working together to sustain life. Figure 1a represents a schematic illustration of a biological mutlicellular structure and its associated componets consisting of living cells (for descriptive purposes, the concept of a generalized eukaryotic cell is introduced) as microconfined regions surrounded by an extracellular matrix (ECM). The cell membrane (composed of a lipid bilayer with embedded protein channels, Fig. 1a, inset) is largely responsible for the cell's unique structure and a variety of cellular processes that are indispensable for life process, including sensory organ homeostasis[13]. Integrins are the proteins that facilitate cell–ECM adhesion and are also involved in the rapid transmission of physiological mechanical stimuli from the ECM to the cell surface[14]. In human skin, the ultrasensitive pressure-sensing capability over a wide spectrum of pressures can be realized by the unique combination of structural and functional features of various mechanoreceptor cells (e.g., Merkel cells, Meissner's corpuscles, Pacinian corpuscles) found in human skin. Although natural skin mechanoreceptors vary in shape and structure, they share some common structural (i.e., existence of plasma membrane, ion channels) and functional (i.e., ion transport) features of eukaryotic cells and they can be represented as eukaryotic cells, in general, for descriptive purposes (Fig. 1b). Under equilibrium, polarized resting membrane potential usually with a negative voltage in the cell interior as compared to the cell exterior ranging from −40 mV to −80 mV is maintained due to the concentration gradient of ions (e.g., $Na^+/K^+$) across the cell membrane, and ion channels (pore-forming membrane proteins) are closed (Fig. 1b, inset). Physical distortion of the mechanoreceptor's cell membrane in response to external mechanical stimuli (Fig. 1c) causes ion channels to open, resulting in the pumping of $Na^+$ ions (Fig. 1c, inset) across the cell membrane to generate an action potential that is sent to the central nervous system as encoding information about the stimulus[13]. In our previous work on visco-poroelastic ionic polymer pump-based mechanotransducer skin, we could successfully mimic the tactile sensing mechanism of Merkel cells[12].

Here, inspired by the structural and the functional features of biological multicellular structure, we demonstrate a synthetic multicellular hybrid ion pump (SMHIP) and explore its potential application in ultrasensitive ionic mechanoreceptor skin. Our rationally designed SMHIP is composed of IL (1-ethyl-3-methyl-limidazoliumbis (trifluoromethyl-sulfonyl)imide ($[EMIM]^+$ $[TFSI]^-$ cation–anion pairs)) confined on silica microstructures (dispersed phase) embedded in thermoplastic polyurethane (TPU) elastomeric matrix (continuous phase), which correspond to the physical analog of living cells and the ECM of biological multicellular structures, respectively. The analogy between our SMHIP and biological cellular structure might be partial because

the living cells in biological cellular structures contain several internal organelles. The key innovation of the presented work is our material design, where a facile chemical bottom-up strategy resulted in situ organization of $[EMIM^+][TFSI^-]$ ion pairs on the surface of silica microspheres in an innovative artificial plasma membrane geometry. Furthermore, the use of silica microstructures as effective ion-confining matrixes in the presented work allowed the dynamic confinement of ionic fluids under external stimuli, which is known to be the key to fabricating ultrasensitive artificial mechanoreceptor skin over a wide spectrum of pressures. The ionic mechanoreceptor skin developed in this work provides an emphatic solution to the low sensitivity, narrow pressure sensing range, and complex device architecture-related issues associated with pressure-sensitive artificial skins developed so far.

## Results

Figure 1d illustrates the conceptual design of our SMHIP, where $[EMIM]^+[TFSI]^-$ ion pairs confined on the surface of silica microstructures embedded in TPU constitute a synthetic multicellular structure with an artificial plasma membrane geometry. Artificial plasma membrane consists of stepwise layers of $TFSI^-$ anions tethered on the surface of silica microspheres via H-bonds with silanol groups (Fig. 1d, inset right), surrounded by $EMIM^+$ cations driven by the Coulomb coupling force with the anions together with the π–π stacking interaction of imidazolium rings (detailed in the molecular characterization section). Some silanol groups of silica–IL structures (artificial cells) in SMHIP are engaged in supramolecular hydrogen-bonding interactions with surrounding TPU matrix (artificial ECM), which directly mimics the biological integrins (proteins that facilitate cell–ECM adhesion and involved in rapid transmission of physiological mechanical stimuli from the ECM to the cell)[14].

Figure 1e illustrates the piezocapacitive, pressure-sensitive ionic mechanoreceptor skin comprising IL–silica–TPU SHMIP film as the pressure-sensing matrix, sandwiched between silver nanowires/polydimethylsiloxane (PDMS) deformable electrodes that reflect the upper layer of human skin. Before stimulus, most of the $[TFSI^-][EMIM^+]$ ion pairs are confined on the surface of the silica microstructures in an artificial plasma-membrane geometry (Fig. 1e, inset), which is due to H-bond–co-coulomb interactions, although π–π stacking interactions between imidazolium rings of $[EMIM^+]$ cations are also observed (detailed in the molecular characterization section). However, few of the ion pairs can exit in the surrounding TPU matrix, mainly through intercalation between TPU hard segments[12]. Under external stimulus (Fig. 1f), SMHIP ionic mechanoreceptor skin engages in ion pumping, which establishes an electric double layer (EDL) at the IL–silica–TPU/electrodes interface, as confirmed by the electrical characterization (Supplementary Note 1 and Supplementary Fig. 1). The main pressure-sensing mechanism in our ionic mechanoreceptor skin should be originated from the pumping of $[EMIM^+][TFSI^-]$ ion pairs from the surface of the silica microstructures (artificial mechanoreceptors) (Fig. 1f) due to the pressure-induced cleavage of H-bonds and/or π–π interactions in SMHIP (Fig. 1f, inset), as claimed in biological systems[15] and various synthetic supramolecular polymeric networks[16–18].

Furthermore, the molecular simulation study of elastomer–silica composites[19] (where the applied stress was mainly accommodated in the vicinity of silica fillers) suggests that the mechanical stress applied to SMHIP is mainly received by ion pairs on the surface of silica microstructures. Non-covalent interactions between silica and $[EMIM^+][TFSI^-]$ ion pairs can act as energy-dissipation sites under mechanical stress[18], via

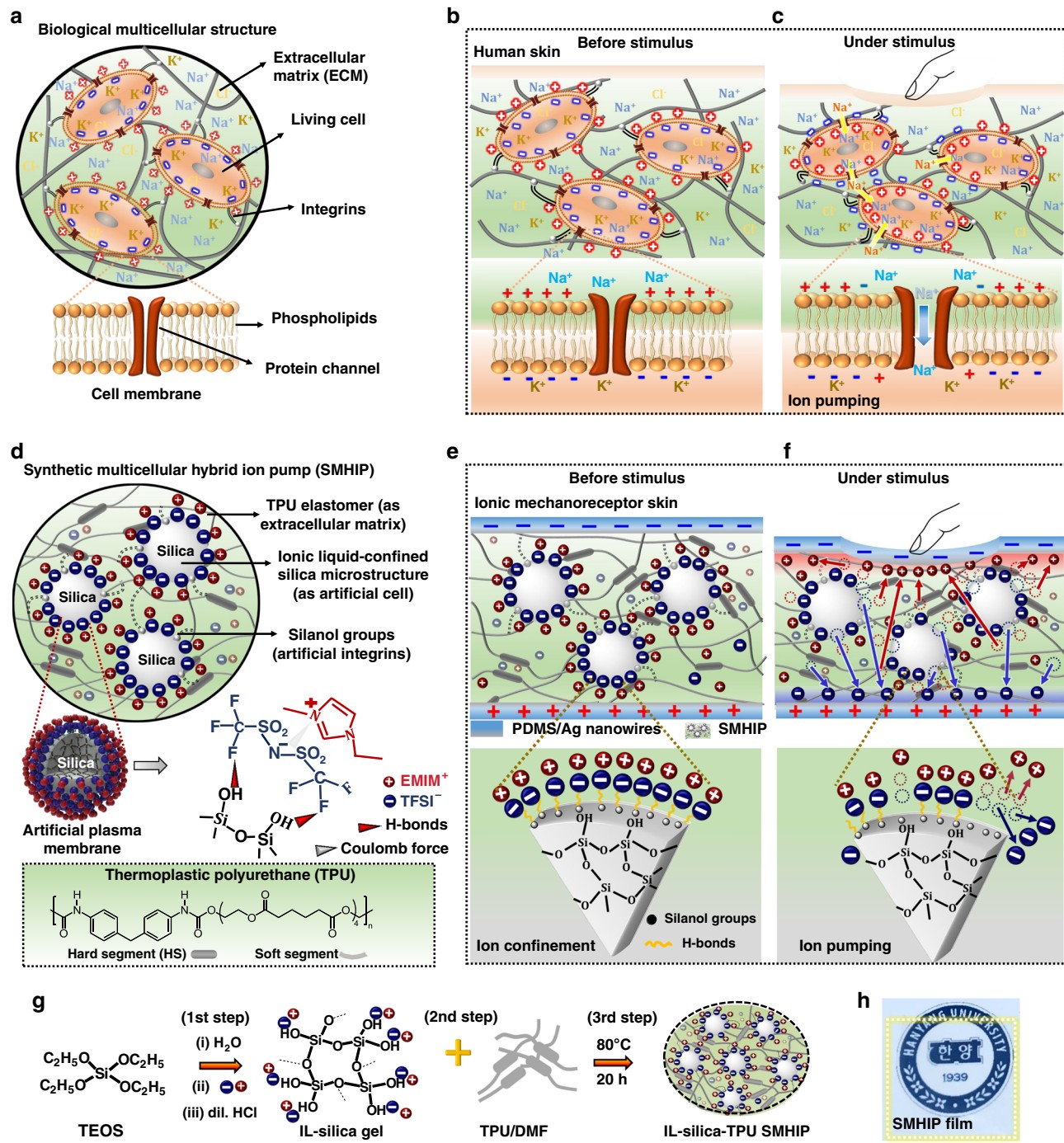

cleavage of silica–TFSI$^-$ H-bonds and π–π interactions of EMIM$^+$ cations, that results in pressure-induced ion pumping in SMHIP ionic mechanoreceptor skin. Electrical characterization of SMHIP films under external stimulus and after removing stimulus conditions (Supplementary Note 1) established the reversible pumping of ions (i.e., recovery of initial state of ions in SMHIP after removal of external stimulus). Breaking and recreation of [TFSI$^-$]–silica H-bonds and π–π stacking interactions of [EMIM$^+$] cations in our SMHIP, similar to previously reported self-healing polymers[18,20] and hydrogels[21,22], should be responsible for the reversible movement of ions in IL–silica–TPU SMHIP-based piezocapacitive devices. Spectroscopic analysis of IL–silica–TPU SMHIP after various external stimuli (will be discussed later) confirms the excellent reversibility of molecular interactions in SMHIP. Figure 1g illustrates the brief schematic of

SMHIP preparation (Supplementary Note 2 and Supplementary Figs. 2, 3) in different steps: (i) synthesis of IL–silica gel through sol-gel process and (ii) blending IL–silica gel and TPU/DMF mixture to obtain transparent tri-component solution (IL–silica–TPU), followed by various heat treatment processes to develop transparent cellular-structured IL–silica–TPU films (Fig. 1h). In this work, 20 wt% is the optimized concentration to obtain the shape- and size-controlled IL-confined silica micro-structures (termed as artificial cells) well dispersed in the TPU polymer matrix (ECM) (Supplementary Note 2 and Supplementary Figs. 4–7).

**Morphological characerization of SMHIP**. The field-emission scanning electron microscope (FE-SEM) image (Fig. 2a) of IL (20

**Fig. 1** Conceptual design of SMHIP and ionic mechanoreceptor skin. **a** Schematic of a biological multicellular structure. Inset is the close view of cell membrane. **b** Schematic of human skin as a multicellular organ consisting of various mechanoreceptor cells. Inset shows a close view of an individual mechanoreceptor's cell membrane before external stimulus. A polarized resting membrane potential is maintained across the cell membrane due to the concentration gradient of $Na^+$ and $K^+$ ions. **c** Schematic of human skin under deformation. Inset is the detailed view of an individual mechanoreceptor's cell membrane under deformation. Pumping of $Na^+$ ions through ion channels across the cell membrane results in depolarization of mechanoreceptor cell membrane and establishment of action potential. **d** Design of a SMHIP composed of IL ([$EMIM^+$][$TFSI^-$], cation–anion pairs) confined on the surface of silica microstructures (as artificial cells) dispersed in TPU elastomer composed of hard and soft segments (as extracellular matrix). Silica surface silanol groups are engaged in H-bonding interactions (black dotted lines) with TPU polymer chains, which directly mimic the biological integrins. Inset on the left is the schematic illustration of an artificial plasma membrane consisting of stepwise layers of $TFSI^-$ anions tethered on the surface of silica microspheres via H-bonds with silanol groups (inset right), surrounded by $EMIM^+$ cations driven by the Coulomb force with the anions. Lowermost inset is the schematic of chemical and segmented representation of TPU. **e** Design of piezocapacitive ionic skin composed of SMHIP film (IL–silica microstructures as artificial mechanoreceptor cells) sandwiched between silver nanowires/PDMS flexible electrodes with a voltage of 1 mV to 1 V. Inset is the close view of artificial mechanoreceptor's plasma membrane under equilibrium. Ionic species are confined via H-bonds. **f** Schematic of ionic mechanoreceptor skin under deformation. Inset is the close view of artificial mechanoreceptor's plasma membrane. Pumping of ions from the surface of silica microstructure due to pressure induced breaking of H-bonds (between $TFSI^-$ and silanol groups) and establishment of EDL at SMHIP/electrode interfaces. **g** Brief schematic of SMHIP preparation. **h** Photo image of SMHIP film (2 cm × 2 cm, yellow box) placed on HANYANG UNIVERSITY logo, indicating high transparency

wt%)–silica–TPU (i(20)–silica–TPU) SMHIP represents the phase-separated morphology; monodispersed silica microspheres (<Diameter> = $7.2 \pm 1.5\,\mu m$) are well dispersed in SMHIP. Energy-dispersive X-ray (EDX) spectroscopy elemental maps (Fig. 2a) and line scan profiles (Fig. 2b) for C (mainly coming from TPU), Si (coming from silica), F and/or S (coming from $TFSI^-$), and N (coming from $EMIM^+$, $TFSI^-$, and TPU collectively) unambiguously support the multicellular morphology of SMHIP. In SMHIP, the dispersed phase consists of silica–[$TFSI^-$] [$EMIM^+$] (as artificial cells), continuous phase consists of TPU polymer (as artificial ECM) and [$TFSI^-$][$EMIM^+$] serves as cellular fluid, although few of the ion pairs may exist in the TPU matrix. More interestingly, high intensities of F and N in the circumferential and inner regions of silica microstructures, respectively, establish a microscopic artificial plasma membrane structure consisting of ordered stepwise layers of $TFSI^-$ and $EMIM^+$ on silica microspheres (Fig. 2a, inset). As guided by several molecular dynamics simulations and experiments available in literature[23–25], where silica surfaces directed the IL cations and anions to form stacked layers alternately, in our case, H-bonding interaction between $TFSI^-$ and silanol groups in IL–silica gel (as illustrated in Supplementary Fig. 2 and further confirmed by Fourier-transform infrared (FTIR) results shown in Supplementary Fig. 3, for details, see Supplementary Note 2) induces conformational change of the $TFSI^-$ anion[26] during the in situ growth of silica microstructures (in the third step of synthesis procedure, Supplementary Fig. 2), which can align [$EMIM^+$] cations driven by coulomb interactions to form stepwise layers of $TFSI^-$ and $EMIM^+$ on the silica surface. This is reflected in the relatively high intensity of F in the circumferential regions of the silica microstructures, as observed in EDX elemental maps and line scans (Fig. 2b and Supplementary Figs. 5–7). In literature, silica has been extensively used to immobilize ILs in a confined geometry with well-modified structural and dynamic properties of ILs, given that strong interactions of IL cations and anions with silica matrix have led to layering and structural heterogeneity of ILs upon confinement[27,28]. Importantly, nitrogen (N) is the constituent element of $EMIM^+$, $TFSI^-$, and TPU as well, and due to embedded nature of IL-confined silica microstructures in TPU polymer matrix, these species ($EMIM^+$, $TFSI^-$, and TPU) co-exist in the silica region. Therefore, a uniform elemental distribution for N is observed in the silica region, which signifies N signals coming from $EMIM^+$, $TFSI^-$, and TPU collectively. FE-SEM/EDX characterizations of i(20)–silica–TPU SMHIP after several repetitive loading/unloading cycles of a wide range of mechanical forces (Supplementary Fig. 8) glorify the excellent structural integrity of our SMHIP.

Transmission electron microscopy (TEM) and high-resolution TEM images (Fig. 2c) reveal the amorphous nature of silica microstructures.

Raman analysis further verifies the artificial plasma membrane structure of our i(20)–silica–TPU hybrid ion pump. Raman intensity maps (Fig. 2d) for TPU hard segment aromatic stretching[29] at 1616 $cm^{-1}$ and silica Si–O symmetric stretching[30] at 465 $cm^{-1}$ corresponding to the marked area in optical microscopy image illustrate lower intensity (blue) for TPU hard segments and relatively higher intensity (green) of silica in the investigated area (a kind of phase-separated morphology where dispersed silica structures are mainly interacting with TPU soft segments). Intensity variations in Raman maps corresponding to $TFSI^-$ ($SO_2$ and $CF_3$ symmetrical stretching)[31] at 1253 $cm^{-1}$ (red, high intensity at the circumferential region of silica) and $EMIM^+$ imidazolium ring in-plane symmetrical stretching, $CH_3(N)$ stretching, and $CH_2(N)$ stretching[31–33] at 1558 $cm^{-1}$ (red, high intensity in the inner region of silica) affirmed the artificial plasma membrane (layers of $TFSI^-$ and $EMIM^+$ on silica microspheres, Fig. 1d, inset) concept, as established by the EDX elemental analysis above. The lack of observable changes in terms of the relative intensity of the Raman maps for individual species, even after many repetitive loading/unloading cycles of mechanical forces (Fig. 2e), further supports the excellent structural integrity of our SMHIP and even the artificial plasma membrane of IL–silica artificial mechanoreceptors, as also evidenced previously by FE-SEM/EDX (Supplementary Fig. 8).

**Molecular characterization of SMHIP.** Figure 3 shows the Raman spectra of i(20)–silica–TPU SMHIP and i(20)–TPU (IL (20 wt%)–TPU) obtained under identical experimental conditions. To support our hypothesis of confined ionic species ([$EMIM^+$][$TFSI^-$]) on silica microstructures in i(20)–silica–TPU SMHIP, i(20)–TPU is used as reference. A blue shift of the $TFSI^-$ expansion–contraction mode[26,27] at ~740 $cm^{-1}$ (Fig. 3a) in i (20)–silica–TPU can be interpreted as the confinement effect of silica on $TFSI^-$ ions due to H-bonding interactions between $TFSI^-$ and silanol group, as claimed in previously reported silica ionogels[26,34]. H-bonding interactions between $TFSI^-$ and silanol groups can induce a conformational change (Fig. 3a, inset) in TFSI-based ILs at the silica/IL interface[34], which is reflected in the low-frequency Raman spectra (250–360 $cm^{-1}$, sensitive to the conformational change of $TFSI^-$)[35] of i(20)–silica–TPU SMHIP (Fig. 3b). Interestingly, i(20)–silica–TPU SMHIP is dominated by $TFSI^-$ cisoid ($C_1$) conformers (277, 312, and 327 $cm^{-1}$) compared to i(20)–TPU, where $TFSI^-$ transoid ($C_2$) conformers (295,

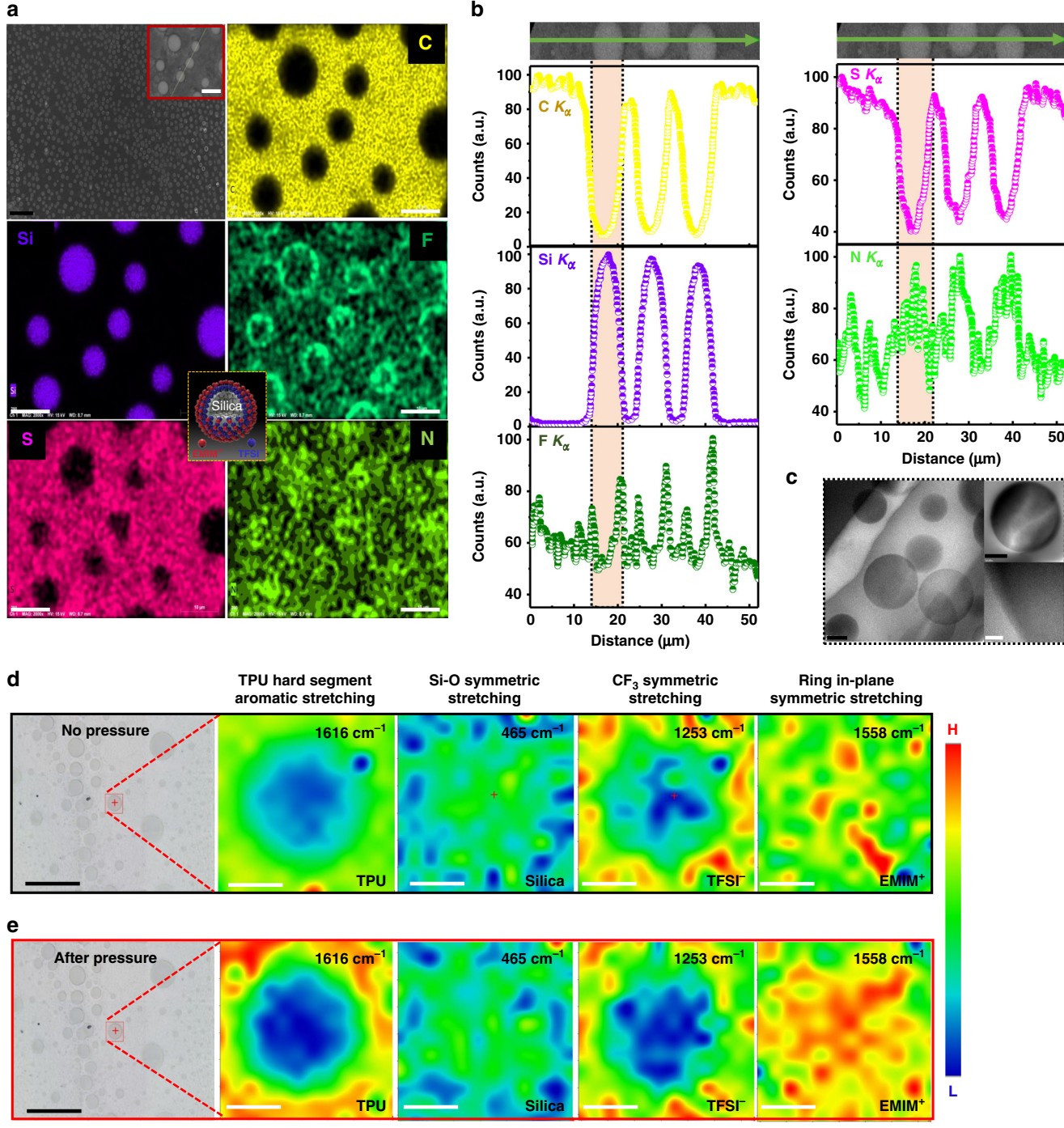

**Fig. 2** Morphological characterization of IL(20 wt%)–silica–TPU SMHIP. **a** FE-SEM image and EDX-elemental maps (corresponding to FE-SEM image shown in inset, red box) for C, Si, F, S, and N. Scale bars, 20 µm (black); 10 µm (white). Inset in the middle (yellow box) illustrates the microscopic artificial plasma membrane structure consisting of ordered stepwise layers of TFSI$^-$ and EMIM$^+$ on silica microspheres. **b** EDX line scan profiles for C ($K_\alpha$), Si ($K_\alpha$), F ($K_\alpha$), S ($K_\alpha$), and N ($K_\alpha$) along the noted path (green arrow, inset). The x-axis represents the distance along the scan line (green arrow) and y-axis represents the relative elemental count along the scan line. Relatively higher intensity of F and S (coming from TFSI$^-$) at the circumferential region of silica structures and relatively high intensity of N (coming from TFSI$^-$, EMIM$^+$, and TPU collectively) at the center of the silica microspheres observed in elemental maps and line scans establish a microscopic artificial plasma membrane structure consisting of ordered stepwise layers of TFSI$^-$ and EMIM$^+$ on silica microspheres (Fig. 2a, middle inset, where gray sphere represents silica microstructures, blue anions represent TFSI$^-$ ions, and red cations represent EMIM$^+$ ions). **c** TEM and HR-TEM images. Scale bars, 0.5 µm (black); 20 nm (white) **d** Raman intensity maps (before external pressure) corresponding to marked area (red box) in optical microscopy image at 1616 cm$^{-1}$ (aromatic stretching of TPU hard segment), 465 cm$^{-1}$ (Si-O symmetric stretching of silica matrix), 1253 cm$^{-1}$ (SO$_2$ asymmetrical stretching and CF$_3$ symmetrical stretching of TFSI$^-$ anion), 1558 cm$^{-1}$ (ring in plane symmetrical stretching, CH$_3$(N) stretching, CH$_2$(N) stretching of EMIM$^+$ cation); the color bar represents Raman intensity profile (H—highest and L—lowest). **e** Raman intensity maps obtained after ten repeated loading/unloading cycles of external pressure applied by thumb on i(20)–silica–TPU SMHIP film. Scale bars, 100 µm (black); 10 µm (white)

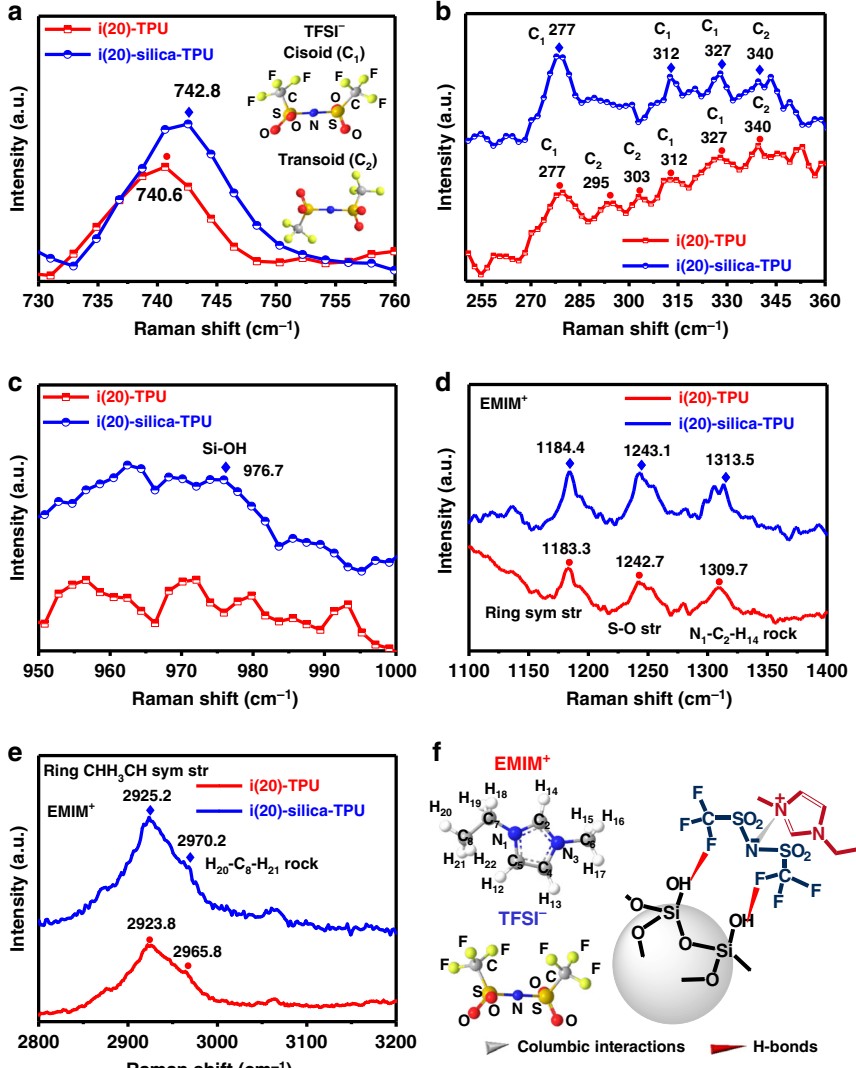

**Fig. 3** Molecular characterization of IL(20 wt%)–silica–TPU SMHIP. **a** The Raman spectra in the spectral range 730–760 cm$^{-1}$ (corresponding to TFSI$^-$ anion expansion–contraction mode). Inset shows the two conformations of TFSI$^-$ anion, the cisoid (or $C_1$) and the transoid (or $C_2$). **b** Raman spectra of TFSI$^-$ conformational sensitive range 250–360 cm$^{-1}$. **c** Si–OH Raman stretching observed at 976.7 cm$^{-1}$ in i(20)–silica–TPU. **d, e** Raman spectra in the spectral ranges 1100–1400 cm$^{-1}$ and 2800–3200 cm$^{-1}$ (corresponding to EMIM$^+$ vibrational bands). **f** Schematic of molecular structures of [EMIM$^+$] (with its atomic number scheme used in **d, e**), [TFSI$^-$] (with its atomic labeling), and the nature of interactions between silica and [EMIM$^+$][TFSI$^-$] ion pairs in i(20)–silica–TPU SMHIP. Source data for **a-e** are provided as a Source data file

303, and 340 cm$^{-1}$) are also found to be dominant. An increase in the population of the TFSI$^-$ cisoid conformers is a well-noticed conformational change of TFSI$^-$-based ILs in confined systems[26,34,35]. Additionally, the Si–OH stretching mode at ~976.7 cm$^{-1}$ (Fig. 3c) reveals the presence of silanol groups in SMHIP. Similar to the TFSI$^-$ Raman features, relative blue shifting of EMIM$^+$ Raman vibrational modes (ring symmetric stretching, N$_1$–C$_2$–H$_{14}$ rock vibration and H$_{20}$–C$_8$–H$_{21}$ rock vibration)[36] in i(20)–silica–TPU (Fig. 3d, e) indicate strong confinement of EMIM$^+$ in i(20)–silica–TPU SMHIP. Figure 3f is a schematic illustration of the nature of interactions involved in the confinement of [EMIM$^+$][TFSI$^-$] ion pairs on the surface of silica microstructures in our SMHIP.

The FTIR spectroscopy results (Fig. 4a) further confirm the role of silica microstructures as confining matrixes to immobilize [EMIM$^+$][TFSI$^-$] ion pairs in our SMHIP. Shifting of FTIR vibrational bands of SO$_2$, CF$_3$, and S–N–S groups[31] of TFSI$^-$ (characteristic region 1400–1000 cm$^{-1}$, Fig. 4a) toward lower wavenumbers clearly indicates the immobilization of TFSI$^-$ ions

over silica microstructures due to H-bond interactions with silica, as reported previously[26,37]. More preferentially, CF$_3$ groups of TFSI$^-$ can form H-bond interactions with the silanol groups of silica, which is reflected as a relatively large shift of CF$_3$ stretching toward the lower-wavenumber side (Fig. 4a). Similarly, shifting of EMIM$^+$ vibrational bands[37] (C–H symmetric/asymmetric stretching, ring NC(H)NCH stretching, and CH$_3$(N)HCH stretching in the characteristic region 3200–3050 cm$^{-1}$, Fig. 4a) in i(20)–silica–TPU SMHIP toward lower wavenumbers (FTIR spectra have been deconvoluted for clear peak assignments) reveals the existence of π–π stacking interactions of EMIM$^+$ imidazolium rings, as observed in the previously reported polymer–silica–IL nanocomposites[38] and silica–IL assemblies[39,40] prepared under acidic conditions similar to our case (see Methods).

Figure 4b shows the FTIR spectra in the regions 3700–3200 cm$^{-1}$ (assigned to N–H stretching region of TPU, where peak centers at 3335.2 cm$^{-1}$ correspond to H-bonded N–H groups) and 1760–1660 cm$^{-1}$ (assigned to C=O stretching region of TPU,

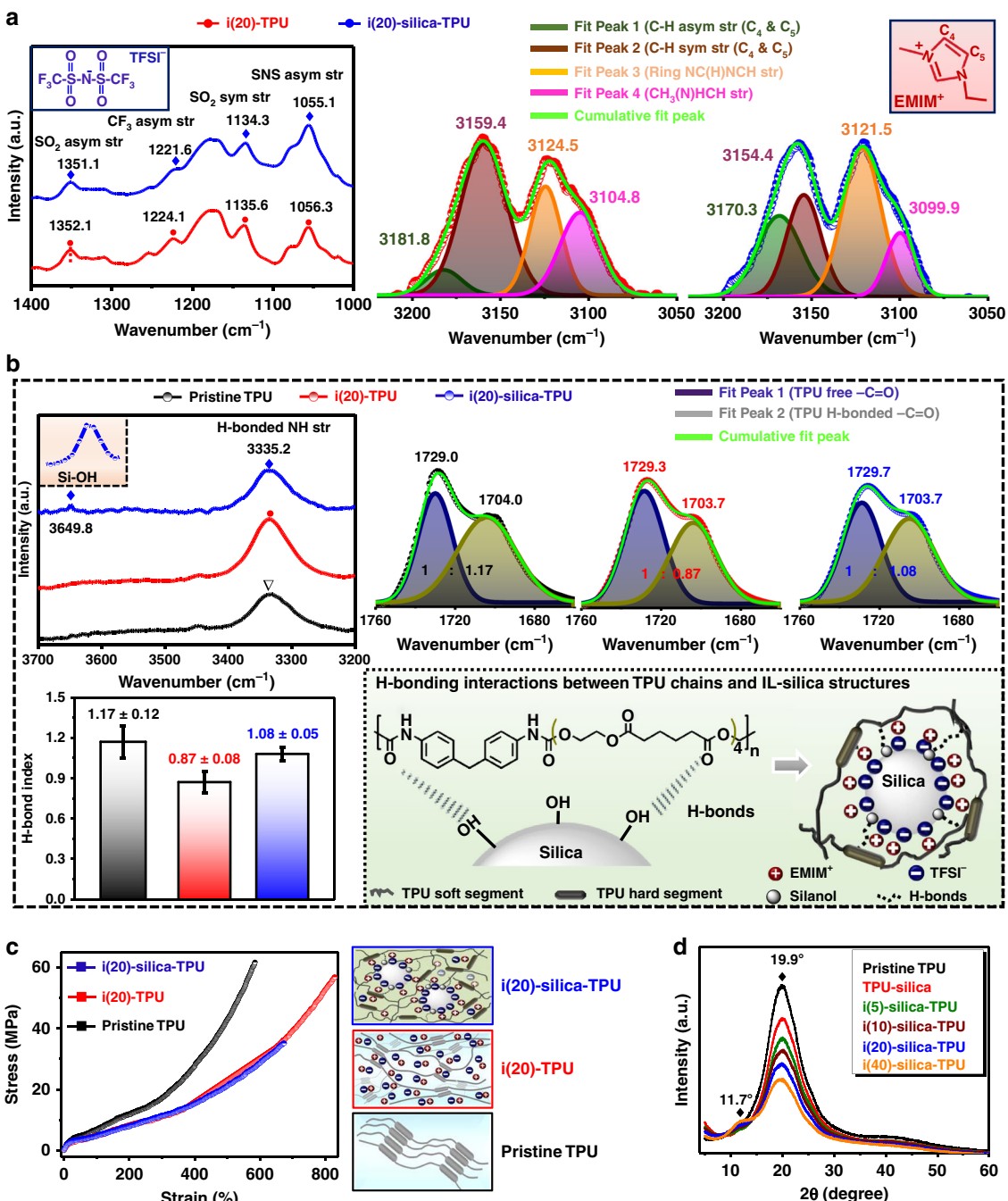

**Fig. 4** Double H-bond networks in SMHIP. **a** ATR-FTIR spectra in the spectral regions 1400–1000 cm$^{-1}$ (corresponding to TFSI$^-$ stretching) and 3200–3050 cm$^{-1}$ (corresponding to EMIM$^+$ stretching); results are shown for i(20)-silica-TPU SMHIP, ristine TPU, and i(20)-TPU. **b** ATR-FTIR spectra in the spectral regions 3700–3200 cm$^{-1}$ (N–H stretching of TPU hard segment) and 1760–1660 cm$^{-1}$ (C=O stretching of TPU). Inset (shown by black dotted box) shows the close-up of Si–OH stretching region observed in i(20)-silica-TPU SMHIP centers at ~3649.8 cm$^{-1}$. Lower inset on the left represents the ca. H-bond index in pristine TPU, i(20)–TPU, and i(20)–silica–TPU SMHIP. Lower inset on the right is the schematic illustration of H-bonding interactions (shown by dotted black lines) between IL–silica (artificial cell) and TPU polymer chains (extracellular matrix) in i(20)–silica–TPU SMHIP. Error bars represent standard deviation in $n = 5$ samples. **c** Stress–strain curves of pristine TPU, i(20)–TPU, and i(20)–silica–TPU SMHIP; schematic illustrates the structural differences. **d** XRD patterns of IL–silica–TPU SMHIP films with varied amount of ILs (5–40 wt%). Source data for **d** are provided as a Source data file

where the two peaks at ~1729 cm$^{-1}$ and ~1703 cm$^{-1}$ correspond to the stretching vibration of free C=O and hydrogen-bonded C=O groups)[41]. The Si–OH stretching vibration mode observed at 3649.8 cm$^{-1}$ (Fig. 4b, inset) in i(20)–silica–TPU signifies the existence of perturbed silanol groups[42], which comes from the interactions of silanol groups with ionic species and/or TPU chains.

A pronounced increase in ca. H-bond index (ratio of the peak area of H-bonded C=O groups/free C=O groups of TPU, Fig. 4b, lower inset left) in i(20)–silica–TPU SMHIP compared to i(20)–TPU reflects the existence of new H-bonds between the silica surface silanol groups and C=O groups of TPU chains (schematic, Fig. 4b, lower inset right), as claimed about the previously reported silicate/

TPU composite[43]. FTIR investigations of TPU–silica and TPU–tetraethyl orthosilicate (TEOS) composites are represented in Supplementary Fig. 9, which signify the crucial role of silanol groups in H-bonds interactions with TPU chains. The H-bonded interfacial interactions between TPU matrix and IL–silica microstructures in SMHIP facilitate the uniform dispersion of IL–silica microstructures in TPU elastomeric matrix (reflected in FE-SEM, Fig. 2a and Supplementary Fig. 4). In order to support our hypothesis of breaking of silica–TFSI⁻ H-bonds under external deformation, FTIR analysis of i(20)-silica-TPU SMHIP film under uniaxial stretching is given in Supplementary Note 3 and Supplementary Fig. 10. Shifting of vibrational bands of TFSI⁻ (Supplementary Fig. 10a) ($SO_2$, $CF_3$, and S–N–S groups) toward higher wavenumbers reflect the weakening/breaking of silica–TFSI H-bonds under external deformation. The shifting of TFSI⁻ vibrational bands toward higher frequencies is a reflection of bulk-like dynamics[34,37]. Similarly, the vibrational bands of EMIM (C–H symmetric/asymmetric stretching, ring NC(H)NCH stretching, and $CH_3$(N)HCH stretching) are shifted to higher wavenumbers under strain. These results are in excellent agreement with the previous studies dealing with spectroscopic investigations of polymeric materials under external deformations, given that the perturbations to the polymeric matrix caused by external deformations (pressure and strain) are reflected by the shifts of the IR bands[44,45]. Moreover, a pronounced decrease of ~38.4% in ca. H-bond index in i(20)–silica–TPU SMHIP (Supplementary Fig. 10b) under strain clearly reflects the breaking of H-bonds networks in SMHIP polymer under external deformation.

No observable changes in the FTIR results were obtained after applying a wide range of pressures (50 kPa to 19 MPa) to i(20)–silica–TPU SMHIP (Supplementary Fig. 11), which confirms the structural integrity and even the reversibility of molecular interactions in our SMHIP. A marginal change in the H-bond index (~0.11%) (Supplementary Fig. 11) even after applying a high pressure of ~19 MPa confirms the reversibility of the H-bonded double network in SMHIP. The double network of H-bonds (H-bonds assisted confinement of ionic fluids on silica surface as well as H-bonds interactions between IL–silica microstructures and TPU chains) in SMHIP can be beneficial for the excellent functioning of SMHIP-based devices under various mechanical loading–unloading processes (Supplementary Note 1), as also claimed in previously reported ionically and H-bonded double-network tough hydrogels[20,46] and polymers[47] with excellent shape-regeneration and healing properties. The double network of H-bonds in i(20)–silica–TPU SMHIP is further reflected in its lower elongation at the break ($\varepsilon$ (%) = 602.2 ± 11.5, Fig. 4c and Supplementary Table 1) and higher Young's modulus ($Y$ = 16.4 ± 1.5 MPa) compared to i(20)–TPU, where the plasticizing effect of ions (schematic, Fig. 4c, inset right) between TPU hard segments[12] is solely responsible for the facile slippage/movement of polymer chains, which yield a lower Young's modulus ($Y$ = 10.4 ± 1.4 MPa) and higher elongation at the break ($\varepsilon$ (%) = 832.8 ± 15.6). The lack of silica characteristics peaks in the X-ray powder diffraction curves (Fig. 4d) of IL–silica–TPU SMHIP films (5–40 wt% of IL content) suggests the amorphous nature of silica structures. The increase in the intensity of low angle peak ($2\theta = 11.7°$) with increasing IL content (pronounced for 40 wt% IL) comes from the intercalation of ions between TPU hard segments[12] with increasing IL concentration as observed in IL–TPU ionic polymers (Supplementary Fig. 12).

**Ultrasensitive mechanotransduction and applications of SMHIP**. Figure 5a shows mechanotransduction of IL–silica–TPU SMHIP films in piezocapacitive pressure-sensor device architecture (ITO/IL–silica–TPU/ITO, film thickness ~170 μm, film area 0.7 cm²) under 1 mV applied bias (@100 Hz) as a function of IL concentration (5–40 wt%). We used indium tin oxide (ITO) glass as a model electrode to exclude the effects of the change in interfacial contact area with IL–silica–TPU films and deformed electrodes in order to prove the concept of confined ionic systems for ultra-sensitive pressure sensing capabilities over a wide spectrum of pressures. A relatively high relative capacitance change $\left(\frac{\Delta C}{C_0} = \frac{C_P - C_0}{C_0}\right)$ (where $C_P$ and $C_0$ denote the capacitance values with and without applied pressure, respectively) in IL–silica–TPU pressure sensors with 10, 20, and 40 wt% of IL (Fig. 5a) demonstrate excellent pressure-sensing capability of IL–silica–TPU SMHIP films over IL (20 wt%)–TPU (without silica) with best achieved in i(20)–silica–TPU. Importantly, the pressure sensitivity $\left(S = \frac{\delta(\Delta C/C_0)}{\delta P}\right)$ (where $P$ denotes the applied pressure) of i(20)–silica–TPU SMHIP film ($S$ = 48.1 – 5.77 kPa⁻¹) is significantly higher (36–48 times) than that of i(20)–TPU film ($S$ = 1.33 – 0.12 kPa⁻¹) over a wide spectrum of pressures (Fig. 5b and Supplementary Tables 2, 3). We attribute this enhanced pressure sensitivity of i(20)–silica–TPU pressure sensor to its tremendously high value of $C_P/C_0$ (~1049), as the sensitivity is directly proportional to $C_P/C_0$.

As shown in Fig. 5c and Supplementary Table 4, exceptionally low $C_0$ in IL–silica–TPU SMHIP pressure sensors, more specifically, in i(20)–silica–TPU ($C_0$ = 49.2 ± 3.7 pF), which is 130 times lower than that of i(20)–TPU ($C_0$ = 6.42 ± 0.44 nF), stems from silica-induced effective confinement of [EMIM⁺] [TFSI⁻] ion pairs under no-pressure condition. In i(20)–silica–TPU SMHIP, the silica microstructures serve as containers to confine [EMIM⁺][TFSI⁻] ion pairs in a micro-structured artificial plasma membrane geometry (as established by various material characterization techniques, Fig. 2) through the H-bond-co-π–π mechanism (as evaluated by FTIR spectroscopy) that generates ultra-low initial capacitance ($C_0$ = 49.2 ± 3.7 pF) of the i(20)–silica–TPU piezocapacitive device, completely different from i(20)–TPU (without silica) where most of the ions are free to establish relatively ultra-high EDL capacitance ($C_0$ = 6.42 ± 0.44 nF), even under no external pressure, as explained in our previous work[12]. The gradual increase in $\frac{\Delta C}{C_0}$ in IL–silica–TPU pressure sensors (Fig. 5a) with increasing pressure and the achieved ultra-high $C_P$ at a pressure of ~135 kPa (Fig. 5c), can be explained by continuous pumping of ions from IL–silica structures due to the pressure/deformation-induced breaking of TFSI-silica H-bonds in SMHIP (Supplementary Note 3) and strengthening of EDL at IL–silica–TPU/electrode interfaces (as shown in Fig. 1f). The frequency dependence of the capacitance of the SMHIP piezocapacitive devices with varying amount of IL content (5–40 wt%) (Supplementary Fig. 13) clearly indicate the EDL phenomena in the devices under pressure.

Electrochemical impedance spectroscopy Nyquist plots (Fig. 5d) of i(20)–TPU (used as reference) and i(20)–silica–TPU SMHIP piezocapacitive devices under no pressure (NP), under pressure (UP), and after removing pressure (AP) indicate the excellent recovery of the initial state in i(20)–silica–TPU SMHIP after the removal of external stimulus, which clearly signifies the reversible pumping of ions in the i(20)–silica–TPU film after the removal of external force (Supplementary Note 1). The reversible movement of ions in SMHIP is further reflected in excellent reproducibility of the i(20)–silica–TPU pressure sensor over a wide range of dynamic mechanical stimuli (1.5–100 kPa), (Supplementary Fig. 14a). Durability tests (Supplementary Fig. 14b) performed on SMHIP pressure sensor signify the excellent stability and structural integrity of SMHIP as a pressure-sensing matrix. SMHIP pressure sensor exhibits a fast response of 60 ms

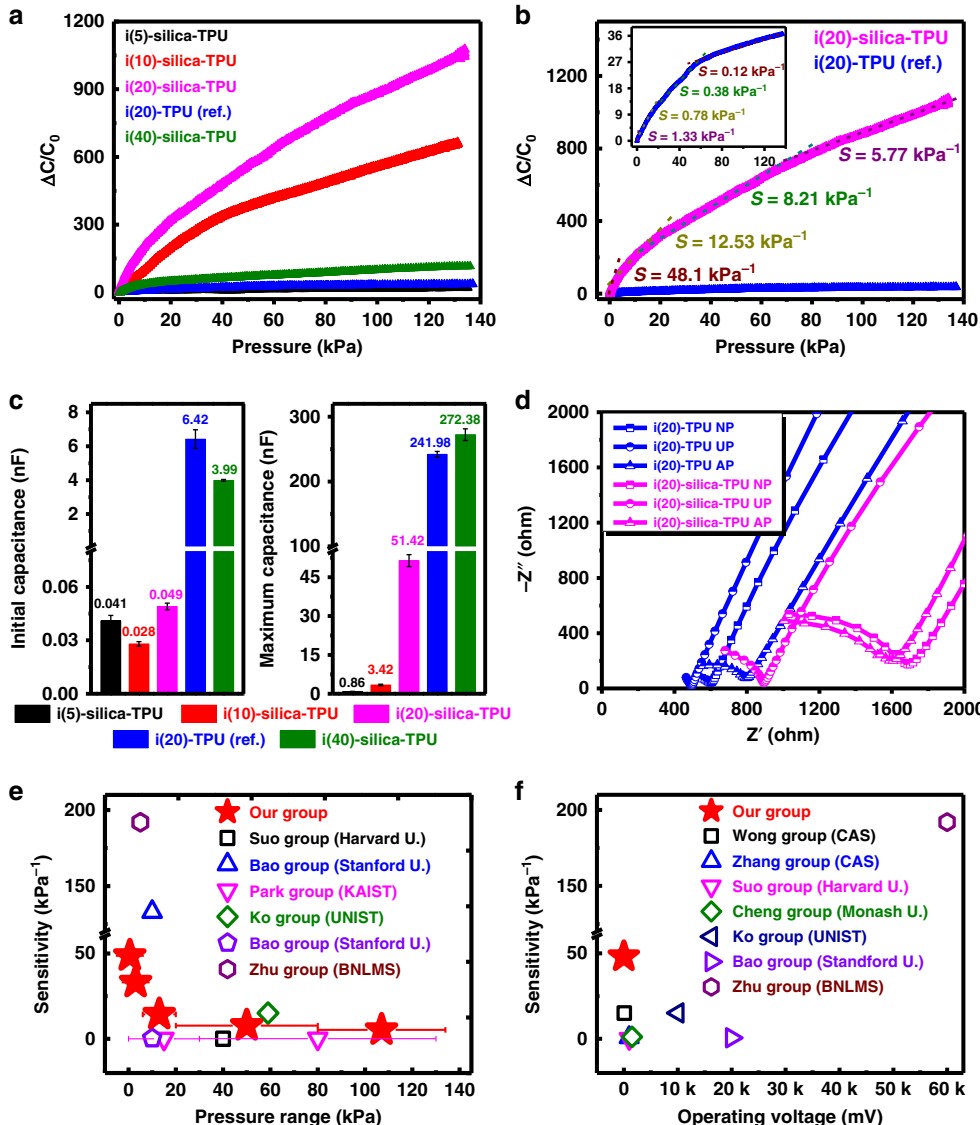

**Fig. 5** Ultrasensitive mechanotransduction in IL–silica–TPU SMHIP. **a** Pressure response of IL–silica–TPU SMHIP pressure sensors under static pressure conditions (1 mV applied bias @ 100 Hz) as a function of IL concentration (5–40 wt%). **b** Comparison of pressure sensitivities of i(20)–silica–TPU SMHIP and i(20)–TPU (inset, blue) (used as reference). **c** Bar graphs represent initial ($C_0$, under no external pressure) and maximum capacitance ($C_p$, under the external pressure of ~135 kPa) of IL–silica–TPU SMHIP piezocapacitive devices (1 mV applied bias @ 100 Hz) as a function of IL concentration (5–40 wt%). For details, see Supplementary Table 4. **d** Impedance Nyquist plots (imaginary part $-Z''$ as a function of real part $Z'$) of i(20)–TPU ionic polymer and i(20)–silica–TPU SMHIP piezocapacitive devices for no pressure (NP), under pressure (UP), and after removing pressure (AP). **e, f** Performance comparison with previously reported pressure sensors. Source data for **d** are provided as a Source data file

and a reset time of 70 ms (Supplementary Fig. 14c). The details of the pressure response of IL–silica–TPU SMHIP films (as a function of IL concentrations) under various experimental conditions can be found (Supplementary Note 4, Supplementary Figs. 15–19, and Supplementary Tables 5–7).

Overall, as depicted in Fig. 5e, our pressure sensor is capable of maintaining high sensitivity (48.1–5.7 kPa$^{-1}$) over a wider range of pressures (0–135 kPa,) than most pressure sensors (Supplementary Table 8) reported to date[1,8–10,48–55], even under an ultra-low operating voltage of 1 mV (10$^2$–10$^5$ times smaller than previous reports[1,8–10,48,49,53–60], Fig. 5f and Supplementary Table 9). Recently, Lee et al.[61] reported a multilayer piezoresistive pressure sensor comprising poly(vinylidene fluoride) (PVDF) and reduced graphene oxide (rGO) microdomed structures with a high pressure sensitivity of 47.7 kPa$^{-1}$ over a pressure range of 0.0013–353 kPa, but the intrinsic limitations of the required high

operating voltage in piezoresistive devices[49,62] limits its use in low-power devices. Figure 6a (bar graphs in red illustrating capacitive change over a wide range of pressures) represents the pressure response of the IL–silica–TPU artificial mechanoreceptor ionic skin sensor array of 9 pixels attached to a model hand (Supplementary Fig. 20a) over a wide range of touch sensations. The ultrasensitive mechanotransduction of our on-model ionic skin sensor array (Supplementary Movie 1), including emotional and discriminative touch signifies that various mechanoreceptors (Merkel cells, Meissner's corpuscles, Pacinian corpuscles) found in human skin that are specialized to sense different pressure regimes (Fig. 6a), can be successfully replaced by only one type of IL–silica–TPU artificial mechanoreceptors in advanced prosthetics. Photographs (Supplementary Fig. 20a, inset) demonstrate the excellent stretchability and rollability of our IL–silica–TPU ionic skin sensor array for applications in wearable electronics. To

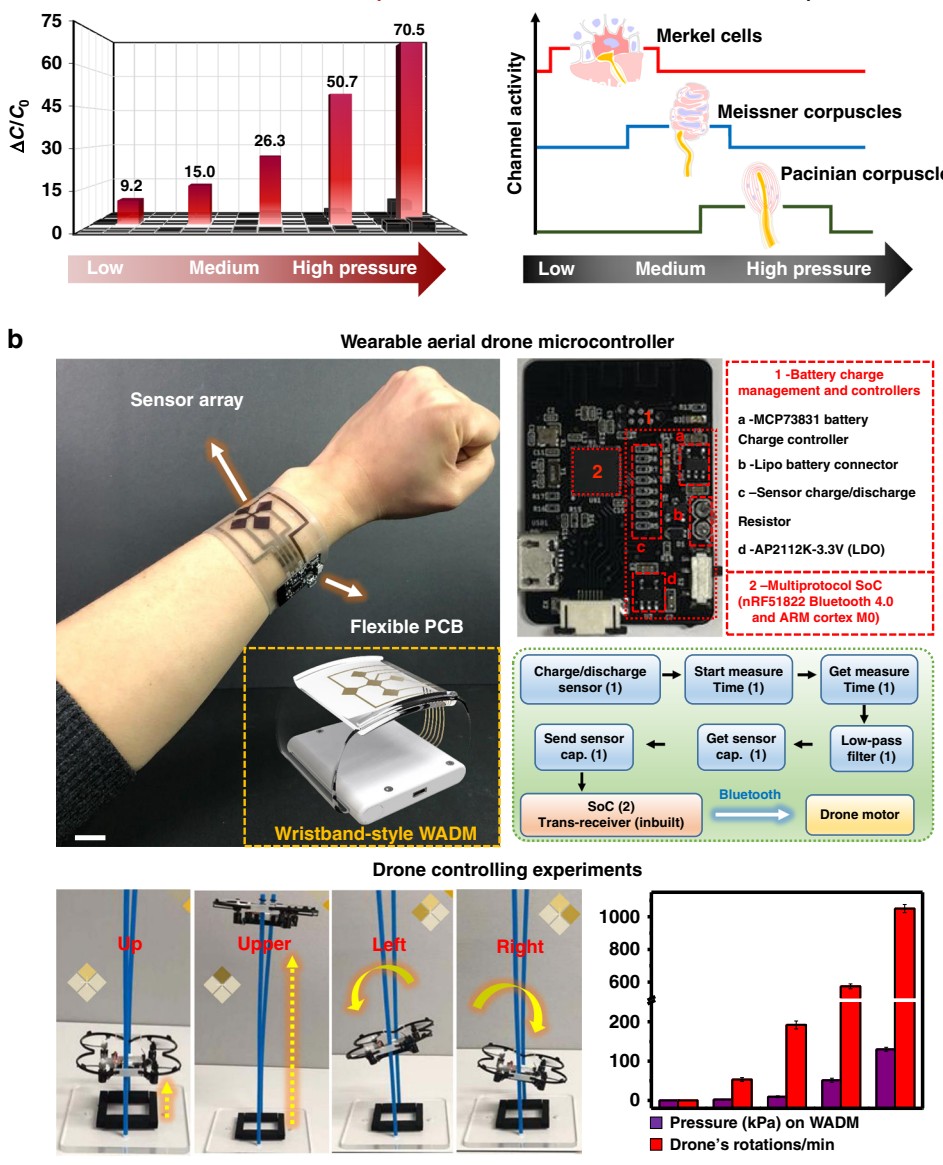

**Fig. 6** Practical applications of SMHIP pressure-sensitive ionic mechanoreceptor skin. **a** Comparison of SMHIP ionic skin mechanoreceptors and human skin mechanoreceptors over a wide range of touch sensations. **b** Photograph of a WADM (fabricated by integrating the IL–silica–TPU flexible pressure sensor array of 4 pixels and flexible wireless PCB) attached to one of the authors' wrist. Inset shown in the yellow dotted box demonstrates a wristband-style WADM based on our SMHIP for commercial use (for details, see Supplementary Fig. 23). Scale bar, 2 cm. Upper inset is the photograph of commercially available flexible wireless PCB with different integrated circuit components labeling (right, red dotted box). System-level functional block diagram (represented by green dotted box) of WADM shows different steps of signal processing and wireless transmission features (for details, see Supplementary Fig. 21) enabling aerial drone control; numbers in parentheses correspond to the labeled components in PCB. Lower inset shows the photographs of real-time experiments of aerial drone control, which include controlling the drone's height, rotation speed, and direction using the WADM developed in this work. Bar curve shows a qualitative response of drone's RPM to the pressure applied on the sensor array of WADM

demonstrate the clinical application of our SMHIP wearable pressure sensor, the sensor was attached to the author's (25 years) neck (Supplementary Fig. 20b) to record the carotid artery blood pressure waves. A typical characteristic carotid artery waveform containing percussion wave ($P1$) and tidal wave ($P2$) was obtained (Supplementary Fig. 20b, inset). The calculated values of radial augmentation index, $AI_r = P2/P1$ (~$0.53 \pm 0.08$) and time delay between the P1-wave and P2-wave, $\Delta T_{DVP} = t_2 - t_1$ ($0.45 \pm 0.02$ s) (the two commonly used parameters for arterial stiffness diagnosis) are in excellent agreements with a healthy person in their mid-twenties as reported by Millasseau et al. via photoplethysmography[63].

The combination of ultra-high pressure sensitivity, large dynamic pressure-sensing range, and excellent stretchability/conformability of our IL–silica–TPU pressure-sensitive ionic mechanoreceptor skin allowed us to fabricate a wearable aerial drone microcontroller (WADM), capable of controlling a drone's number of rotations and even its direction simultaneously and/or selectively during its flight, a completely cutting-edge technological application. A drone is a kind of unmanned aerial vehicle (UAV) that is used for a variety of practical applications, such as military surveillance, commercial purposes, aerial photography, and search and rescue. Figure 6b shows a photograph of a WADM attached to one of the authors' wrist, fabricated via the

integration of SMHIP pressure sensor array of 4 pixels and commercially available integrated circuit components consolidated on a flexible printed circuit board (PCB). The sensor array was fabricated on a mechanically flexible polyethylene terephthalate (PET) film (~30 μm) using Cr (10 nm)/Au (40 nm) electrode patterns (see Methods). A flexible PCB delivers robust signal conditioning, processing, and wireless transmission features using readily available integrated circuit components. MCP73831 microchip (Fig. 6b, upper inset) is a highly advanced linear charge-management controller, which, in combination with AP2112 low-dropout (LDO) regulator, provides a regulated output voltage. Multi-protocol system on chip (SoC) (nRF51822 Bluetooth 4.0 and ARM Cortex M0) facilitates wireless data transmission from the WADM to the drone. A system-level functional block diagram of WADM (represented by green box) shows the signal-conditioning, signal-processing, and wireless transmission paths from the sensor to the custom-developed mobile application and drone motor (for details, see Supplementary Fig. 21). The signal-conditioning/processing path from sensor is initiated by charging/discharging of sensor to read charging/discharging times as output signals as a function of applied pressure or in other words as a function of pressure-induced capacitance change of the corresponding pixel of the pressure sensor array. The built-in microprocessor on SoC relays the conditioned signals to the transceiver for wireless data transmission, enabling successful control of the aerial drone in the presented work. The circuit diagram of the signal-conditioning circuit of WADM is shown (Supplementary Fig. 22). The Fig. 6b inset (yellow dotted box) demonstrates a wristband-style WADM (for details, see Supplementary Fig. 23 and Supplementary Movie 2) based on our SMHIP for commercial use. Photographs of real-time experiments of the aerial drone controlled with a WADM (Fig. 6b, lower inset right) demonstrate an unexcelled level of simultaneous and selective control of the drone's height, number of rotations, and even direction with the wearable drone microcontroller developed in this work (for details, see Supplementary Movies 3 and 4). The bar curve shows a real-time qualitative response of the drone's rotations per minute (RPM) to the applied pressure on the sensor array of WADM.

## Discussion

We have demonstrated a double H-bond network SMHIP that emulates the structural as well as the functional features of biological multicellular structures to fabricate ultrasensitive ionic mechanoreceptor skin. Reversible H-bond triggered ion pumping in SMHIP is the key to maintain ultrahigh pressure sensitivity over a wide spectrum of pressures, compared to previous electronic and ionic skin reports. This ionic mechanoreceptor skin can be integrated over large areas for potential applications in next-generation prosthetic devices. Furthermore, our ultra-sensitive ionic mechanoreceptor skin realizes the fabrication of a wearable drone microcontroller, capable of controlling direction and speed simultaneously and selectively in aerial drone flight. We believe that the research outcomes of the presented work are likely to revolutionize various fields including artificial intelligence, next-generation prosthetics, and human–machine interfaces. Furthermore, the presented work represents a step toward next-generation artificial skin and frontier materials for synthetic biological systems.

## Methods

**Preparation of SMHIP**. Preparation of IL–silica–TPU SMHIP films involves three main steps: (i) preparation of IL–silica gel through sol-gel; (ii) preparation of TPU precursor gel; (iii) preparation of IL–silica–TPU tri-component solution, followed by an optimized heat-treatment process to develop IL–silica–TPU SMHIP films.

In the first step, IL–silica gel was prepared through the sol-gel method. 1-Ethyl-3-methylimidazolium bis(trifluoromethylsulfonyl)imide ($[EMIM^+][TFSI^-]$) and TEOS were purchased from Sigma-Aldrich. In a typical optimized synthesis procedure (e.g., for 20 wt% IL–silica–TPU SMHIP), 0.5 ml of TEOS was added to 0.25 ml of water under continuous stirring at 40 °C for 10 min. Then, an appropriate amount of $[EMIM]^+[TFSI]^-$ (0.4 ml) was added dropwise to the TEOS–water mixture and stirred for another 15 min at same temperature. The molar ratio of TEOS to water to IL was 1:6:0.9. Then, 0.05 ml of hydrochloric acid (0.06 M) was added dropwise to this mixture under continuous gentle stirring at 40 °C and the obtained solution was stirred continuously. During the stirring process, the mixture first becomes turbid (in first ~10–15 min of stirring process), indicating the hydrolysis/condensation of TEOS to form silica network and then transforms in to a transparent clear solution (after ~15–20 min), indicating the formation of IL–silica gel through the interaction of ionic species with silica network. The obtained IL–silica transparent gel was further stirred for 15 min at 40 °C and then used for the next step.

In the second step, TPU precursor gel was prepared by dissolving TPU beads (KA-480, Kolon Industries, Inc.) into N,N-dimethylformamide (DMF, Sigma-Aldrich) at a mass ratio of 1:5 under continuous stirring at 80 °C for 3 h.

In the third step, IL–silica gel (obtained in first step) was added to TPU gel (obtained in the second step) dropwise under continuous stirring at 80 °C, and the resulting IL–silica–TPU tri-component gel was stirred at 80 °C for another 20 h. To obtain IL–silica–TPU SMHIP film of the desired thickness, a fixed amount of IL–silica–TPU tri-component gel was poured in a Teflon dish and heat treated at 80 °C for 72 h under optimized conditions (staring from 40 °C with 10 °C per hour temperature ramp). The IL weight percentage reported in this work stands for the weight ratio of IL to IL + TPU. Various IL–silica–TPU films with varying IL content (5, 10, 20, and 40 wt%) were prepared on the basis of the weight ratio of IL and IL + TPU (keeping the silica precursor concentration unchanged) under identical reaction conditions.

TPU–TEOS and TPU–silica composite films were prepared under identical reaction conditions as mentioned above and used as references.

**Fabrication of ionic mechanoreceptor skin sensor array**. The silver nanowire solution (a concentration of 0.25 mg ml$^{-1}$ in isopropyl alcohol) was prepared from the dilution of a silver nanowire suspension (Nanopyxis Corp., 0.5 wt% in isopropyl alcohol; diameter and length of the silver nanowires was 32 ± 5 nm and 25 ± 5 μm, respectively). The PDMS substrate was prepared by curing a mixture of a base resin and a crosslinker (10:1 by mass, Dow Corning Corp., Sylgard 184) in a plastic Petri dish. After degassing and curing in an oven at 80 °C for 2 h, the PDMS substrate was cut to the desired size. After a sonication process (~1 h), diluted Ag nanowire solution was spray-coated (SRC-200 VT, E-FLEX Korea, with a nozzle of 0.05 mm, spraying pressure 200 mbar) on patterned PDMS heated at 100 °C followed by annealing at 120 °C for 1 h. IL–silica–TPU pressure-sensitive ionic mechanoreceptor skin was fabricated by sandwiching the IL–silica–TPU SMHIP film (area: 0.6 cm × 0.6 cm, thickness of the film ~170 μm) between two patterned PDMS/Ag nanowires electrodes (area: 0.5 cm × 0.5 cm) to fabricate a 3 × 3 sensing array of 9 pixels (Supplementary Fig. 20a). The silver wires (Nilaco Corp., diameter: 50 μm) were attached on the electrodes for connections with the measuring instrument. For pressure-sensing experiments (Supplementary Movie 1), ionic mechanoreceptor skin sensor array was attached to a model hand with tattoo paper and then connected with an LCR meter (Keysight Technologies, E4980A) and data-acquisition/data-logger switch unit (Keysight Technologies, 34970A) for visualizing the capacitance change in the mapping image. Each channel of the 3 × 3-cell sensor array could be configured independently to measure the capacitance of each cell without interference. The measured capacitance was displayed as a mapping image on custom-made data acquisition software (ATM Corp.). The sensing ability, multi-touch, and responsibility of the 3 × 3 ionic mechanoreceptor skin sensor array were measured under 50 kHz, 50 mV.

For measurement of carotid artery blood pressure (Supplementary Fig. 20b), the sensing device was fabricated on a mechanically flexible PET film (~30 μm) using thermally evaporated Cr (10 nm)/Au (40 nm) electrode patterns by sandwiching IL (20 wt%)–silica–TPU SMHIP film (170 μm, area ~ 1 cm$^2$).

**Fabrication of WADM**. WADM was fabricated by merging the SMHIP flexible pressure sensor array and commercially available integrated circuit components consolidated on a flexible PCB. The SMHIP flexible pressure-sensor array of 4 pixels (area of each pixel ~0.81 cm$^2$) was fabricated on a mechanically flexible PET film (~30 μm) using thermally evaporated Cr (10 nm)/Au (40 nm) electrode patterns by sandwiching IL(20 wt%)–silica–TPU SMHIP film (170–180 μm, area ~ 1 cm$^2$). Flexible ultra-thin silver wires (~50-μm thickness) were used to connect the sensor array and flexible PCB. A small rechargeable lithium-ion battery (3.7 V, 100 mAh) was used as a power source.

**Material characterization**. The surface morphological features of IL–silica–TPU SMHIP films were obtained using field-emission scanning electron microscopy (FE-SEM, JSM-6700F, JEOL) equipped with QUANTAX EDX system for elemental analysis. X-ray diffraction analysis (XRD) was performed using a Bruker D2 Phaser desktop X-ray diffractometer operating at 30 kV and 10 mA with a Cu $K_\alpha$ radiation

source. The diffraction scans were acquired using a 5–60° $2\theta$ range, with an ~200-µm thickness of IL–silica–TPU and IL–TPU films (concentration of IL: 0, 5, 10, 20, 40 wt%). The FTIR spectra were recorded by using Bruker Optics GmbH (Germany) spectrometer under attenuated total reflection (ATR) mode (ZnSe crystal). Each spectrum, recorded as the average of 64 scans with a resolution of 2 cm$^{-1}$, was collected from 4000 to 450 cm$^{-1}$. Deconvolution of the FTIR peaks was performed by considering peaks Gaussian with a number of iterations to obtain the best-fit Gaussian peak. Raman spectroscopic measurements were performed by using a Thermo Scientific™ DXR™2 Raman microscope and a Thermo Scientific™ DXR™2xi Raman imaging microscope system. The laser power at the sample plane during Raman imaging was 2.0 mW for 532-nm laser excitation with a spot size of ~10 µm$^2$ and a spatial resolution of 3 µm. Transmission electron micrographs of IL–silica–TPU SMHIP were obtained using a Cryo-Transmission Electron Microscope (ModelFEI (Tecnai F20 G2) at an accelerating voltage of 200 kV). Ultrathin specimens (80-µm thickness) were prepared by Reichert Ultracut S cryoultramicrotome system after freezing the specimen at −100 °C using liquid nitrogen. The cryomicrotomed sections were deposited on 400 mesh copper grids prior to TEM analysis. A universal testing machine (UTM QRUTS-S105, QURO) with a 1-kN load cell was employed to measure the mechanical properties of all polymeric films at 25 °C with a stretching speed of 10 mm min$^{-1}$. All the samples were prepared and tested according to ASTM standards (Test Method D 638–02a, specimen type V). Young's modulus ($Y$) was calculated according to the slope of the stress–strain curves (within 0–5% of strain values). Experimental results properties presented in this study are mean ± standard deviation obtained from the five samples prepared and tested identical experimental conditions.

**Electrical characterization**. Electrochemical impedance spectroscopy (EIS) is a powerful technique to study ion-transport phenomena in polymer electrolytes and their interfaces (i.e., electrode–electrolyte interfaces). EIS measurements were performed at room temperature using an electrochemical analyzer PGSTAT302N (Metrohm Autolab) in a frequency range of 0.1 Hz to 100 kHz with a 10-mV AC signal. A coin cell assembly provided by Hohsen Corp. (Japan) allowed us to perform EIS measurements of different polymer films under different experimental conditions (with and without pressure conditions). For EIS measurements, polymer films (~170 µm) were sandwiched between two stainless-steel discs (diameter = 10 mm, used as electrodes) to achieve a piezocapacitive device configuration. All of the impedance spectra were fitted using the appropriate equivalent circuit models built in NOVA software (Metrohm Autolab) to evaluate the bulk resistance ($R_b$) of the devices. The ionic conductivity was calculated from bulk resistance values as follows: $\sigma = \left(\frac{l}{R_b \times A}\right)$, where $\sigma$ is the ionic conductivity, $l$ the thickness of polymer film sandwiched between electrodes, $A$ the area of the electrode, and $R_b$ the bulk resistance obtained from EIS Nyquist plots. Capacitance measurements were performed at room temperature using an Agilent E4980A Precision LCR Meter. Piezocapacitive devices were fabricated by sandwiching the IL–silica–TPU films (film thickness ~ 170 µm, film area 0.7 cm$^2$) between two ITO glass electrodes (surface resistance ~ 10 Ω sq$^{-1}$). The silver wires (Nilaco Corp., diameter: 50 µm) were attached to the electrodes for connections with the measuring instrument.

**Pressure response of SMHIP films**. A custom-built sensor-probe station (Supplementary Fig. 24) with a programmable $xy$- and $z$-axis stage (0.1-µm resolution) equipped with a force gauge (Mark-10, with 0.005-N resolution) was used to study the pressure response of the assembled pressure sensors. The corresponding pressure was calculated by dividing the load with the sensing dimension of the unit device/pixel. The measuring equipment was connected to the customized LabVIEW-based program that can simultaneously record the in situ capacitance change and applied load. The pressure sensitivity of the devices in different pressure regimes was obtained from the slope of the relative change in capacitance versus the pressure.

## Data availability

The authors declare that all relevant data supporting the results of this work are available within the paper and its supplementary information files. The source data underlying main Figs. 3, 4d, 5d and Supplementary Figs. 10, 11, 15–18 are provided as source data files. Additional data are available from the corresponding author upon request.

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

## Acknowledgements

This work was financially supported by the Centre for Advanced Soft-Electronics under the Global Frontier Project (2014M3A6A5060932) and the Basic Science Research Program (2017R1A2B4012819 and 2017R1A5A1015596) of the National Research Foundation of Korea (NRF) funded by the Ministry of Science, ICT.

## Author contributions

D.H.K. supervised the project. V.A., J.S.K., E.J., S.Y.K., J.K., H.C., Y.S.C, and Y.K. conducted the experiments. V.A., J.S.K., E.J., and S.Y.K. performed the material characterization studies. V.A., J.S.K., E.J., and D.H.K. wrote the manuscript. All authors commented on the manuscript.

## Additional information

**Competing interests:** The authors declare no competing interests.

