## [Peer Review File · Nature Communications]

Reviewers' comments:

Reviewer #1 (Remarks to the Author):

The manuscript entitled "A bioinspired, hydrogen bond-triggered ultrasensitive ionic mechanoreceptor skin" submitted by D. H. Kim and coworkers reports a novel bioinspired ionic polymer composite and its application towards highly pressure-sensitive artificial skin. The close emulation of biological cellular structures through confined ion strategy in this work has resulted in unprecedented pressure sensitivity of piezocapacitive ionic sensor skin over a wide spectrum of pressures, that has been never reported previously. Also the developed material/sensor is utilized for the fabrication of a wearable drone microcontroller, which represents a new application in the field of electronic skins. So, I strongly recommend this work for publication in Nature Communications after the authors address following questions:

1. The authors mentioned that the pressure induced ion pumping in cellular-structured ionic polymer composites and generation of electric double layer phenomena in the piezocapacitive devices (with varied IL contents). However, the data (capacitance vs frequency plot) provided in the manuscript belongs to only IL 20 wt.%. In order to support EDL mechanism under pressure in other materials (with varied IL contents), the authors should provide the capacitance vs frequency behaviors of other IL-silica-TPU composites (with varied amount of IL contents).
2. The authors mentioned about the double H-bonding interactions (between IL and silica and silica and TPU) in IL-silica-TPU ionic composites. What about the molecular interactions in pristine TPU-silica and TPU-TEOS systems in terms of H-bond index? This is not clear in the manuscript.
3. As mentioned by the authors, the breaking of H-bonds in IL-silica-TPU films under pressure is responsible for pressure sensing mechanism of IL-silica-TPU pressure sensor. The manuscript provides spectroscopic data of molecular interactions in no pressure and after pressure conditions. However, the nature of molecular interactions in the material is not studied under pressure. Although the authors provide strong references that describe the breaking of H-bonds in various self-healing polymers under pressure. In my opinion, to support the H-bond breaking mechanism under pressure in this manuscript authors should conduct additional spectroscopic experiments under pressure.
4. Some typo should be corrected.
5. Some other research articles (i.e., J. Mater. Chem. B, 2019,7, 173-197; Adv. Funct. Mater. 2019, 29, 1807343) should be cited.

Reviewer #2 (Remarks to the Author):

Present manuscript entitled "A bioinspired, hydrogen bond-triggered ultrasensitive ionic mechanoreceptor skin" reports on the synthetic multicellular hybrid ion pump (SMHIP) film using ionic liquid (IL) 1-ethyl-3-methylimidazolium bistrifluoromethylsulfoniomide {EMIM}[TFSI] confined on silica microstructures embedded in thermoplastic polyurethane (TPU). The IL-silica-TPU SMHIP film has been developed by synthesizing IL-silica-gel through conventional sol-gel method followed by preparation of thermoplastic polyurethane (TPU) precursor gel and then preparing IL-silica-TPU solution heat treatment process. IL-silica-TPU film with 20 wt% IL resulted optimized film. 20 wt% is the optimized concentration to obtain the shape and size controlled IL-confined silica microstructures (called as artificial mechanoreceptor cells)

Ionic skin sensor array was fabricated by sandwiching the IL-silica-TPU SMHIP film between a couple of patterned PDMS/Ag nanowire electrodes.

The main pressure-sensing mechanism in the hybrid ionic skin was attributed to originated from the pumping of [EMIM][TFSI] ion pairs from surface of silica microstructures due to the pressure induced cleavage of H-bonds and /or pi-pi interactions in SMHIP similar to biological systems.

The present research work may be helpful in synthesizing synthetic ionic mechanoreceptor skin for different applications.

The manuscript can be accepted for publication subject to the following revisions

1. There should be more clarity in the introduction part.
2. Nyquist plots (Fig 5d) should have identical scales for Z' and Z'' axes.
3. Since the main pressure sensing mechanism in the proposed ionic skin has been attributed to the pumping of ionic liquid [EMIM][TFSI] ion pairs from surface of silica microstructures due to pressure induced cleavage of hydrogen bonds/ π - π interactions in SMHIP therefore, some references dealing with the interaction of cations/anions of ionic liquid molecules with the pore-wall surface of silica matrix are missing e.g. doi.org/10.1016/j.pmatsci.2014.03.001, doi.org/10.1039/C4RA04362, doi.org/10.1021/jp2003358.

Reviewer #3 (Remarks to the Author):

The authors have reported "dynamic confinement" of charged molecules in an ionogel by introducing silica particles. Silica particles in the gel are making hydrogen bonding with bis(trifluoromethylsulfonyl)imide(TFSI⁻), so they could drop the conductivity of the ionic liquid. However, as the confinement of TFSI⁻ could be released by a mechanical force, the gels could recover their conductivity when they undergo a mechanical deformation. Those kinds of dynamic capturing and releasing of ions by silica particles are firstly introduced. In that sense, this reviewer think the manuscript has enough novelty for the publication. Furthermore, because the authors have studied the confinements very well with EDX, Raman spectroscopy, and FT-IR, this reviewer think the manuscript is also well organized with sufficient verifications about their hypothesis. Therefore, this reviewer recommends the publication of this manuscript in Nature Communications. However, there are some unclear factors I would like to ask and minor points which should be revised. Please consult the followings;

1. In Fig. 2a, compare to carbon map, nitrogen map shows homogeneous distribution. Any ideas about the nitrogen map?
2. Similar with previous question. In Fig. 2d and 2e, the Raman intensity map of 'ring-in-plane symmetric stretching' is hard to understand. Any ideas about it?
3. EDX line scan profiles at Fig. 2b need y axis legend.

June 26th, 2019

Response to Reviewer #1

Thank you for your invaluable comments.

We revised the manuscript according to your comments.

The manuscript entitled "A bioinspired, hydrogen bond-triggered ultrasensitive ionic mechanoreceptor skin" submitted by D. H. Kim and coworkers reports a novel bioinspired ionic polymer composite and its application towards highly pressure-sensitive artificial skin. The close emulation of biological cellular structures through confined ion strategy in this work has resulted in unprecedented pressure sensitivity of piezocapacitive ionic sensor skin over a wide spectrum of pressures, that has been never reported previously. Also the developed material/sensor is utilized for the fabrication of a wearable drone microcontroller, which represents a new application in the field of electronic skins. So, I strongly recommend this work for publication in Nature Communications after the authors address following questions:

Q1) *The authors mentioned that the pressure induced ion pumping in cellular-structured ionic polymer composites and generation of electric double layer phenomena in the piezocapacitive devices (with varied IL contents). However, the data (capacitance vs frequency plot) provided in the manuscript belongs to only IL 20 wt.%. In order to support EDL mechanism under pressure in other materials (with varied IL contents), the authors should provide the capacitance vs frequency behaviors of other IL-silica-TPU composites (with varied amount of IL contents).*

Response: We appreciate the comment from the reviewer. The capacitance vs frequency plots of IL-silica-composites (5, 10, 40 wt.% of IL) in piezocapacitive device configurations are obtained and added as Supplementary Figures 13a-c in the revised version. All the samples were tested under similar experimental conditions as mentioned in the "Methods" section of the main manuscript. The frequency dependence of the capacitance of the SMHIP piezocapacitive devices with varying amount of IL content (5-40 wt.%) indicates the EDL phenomena in the devices under pressure. We reflected this on page 26, in the revised supplementary information (**Figure S13**).

Supplementary Figure 13. Frequency vs. capacitance plots of C_0 and C_p in various SMHIP piezocapacitive devices with varying amount of IL. **a** i(5)-silica-TPU (@ 100 mV). **b** i(10)-silica-TPU (@ 1 mV). **c** i(40)-silica-TPU (@ 1 mV).

Q2) The authors mentioned about the double H-bonding interactions (between IL and silica and silica and TPU) in IL-silica-TPU ionic composites. What about the molecular interactions in pristine TPU-silica and TPU-TEOS systems in terms of H-bond index? This is not clear in the manuscript.

Response: We thank reviewer for asking us to clarify this point. TPU-silica and TPU-TEOS composite films were prepared under identical conditions (Methods) and characterized using Fourier-transform infrared (FT-IR) spectroscopy technique in order to find out the nature of molecular interactions and hydrogen bond index (H-index) in these composites. Supplementary Figure 9 shows the FT-IR spectra of TPU-silica and TPU-TEOS composites in the spectral region $1760\text{--}1660\text{ cm}^{-1}$ (assigned to C=O stretching of TPU, where the two peaks at $\sim 1729\text{ cm}^{-1}$ and $\sim 1703\text{ cm}^{-1}$ correspond to the stretching vibration of free C=O and hydrogen-bonded C=O groups films). A pronounced increase in *cal.* H-bond index in silica-TPU composite (1.11 ± 0.09) compared to TPU-TEOS (0.77 ± 0.12), signifies the H-bonding interactions between silica and TPU chains due to surface Si-OH groups, as observed in i(20)-silica-TPU SMHIP. We reflected this on page 22 in revised supplementary information (**Figure S9**).

Supplementary Figure 9. FT-IR spectra of TPU-silica and TPU-TEOS composites. ATR-FTIR spectra in the spectral region 1760–1660 cm^{-1} (C=O stretching of TPU). Deconvolution of the FTIR peaks was performed by considering peaks Gaussian with a number of iterations to obtain the best-fit Gaussian peak. The ratio of the peak area of H-bonded -C=O bands (centers at approximately 1703 cm^{-1}) to the free -C=O bands (centers at approximately 1730 cm^{-1}) is termed as H-bond index. Bar graph represents H-bond indices in different composites.

Q3) *As mentioned by the authors, the breaking of H-bonds in IL-silica-TPU films under pressure is responsible for pressure sensing mechanism of IL-silica-TPU pressure sensor. The manuscript provides spectroscopic data of molecular interactions in no pressure and after pressure conditions. However, the nature of molecular interactions in the material is not studied under pressure. Although the authors provide strong references that describe the breaking of H-bonds in various self-healing polymers under pressure. In my opinion, to support the H-bond breaking mechanism under pressure in this manuscript authors should conduct additional spectroscopic experiments under pressure.*

Response: We appreciate the reviewer's comment. In order to study nature of interactions in IL-silica-TPU films under external deformation, FT-IR studies of i(20)-silica-TPU SMHIP film

during uniaxial stretching at ambient conditions were performed. Supplementary Figure 10 shows the FT-IR spectra of i(20)-silica-TPU SMHIP at normal state ($\varepsilon = 0\%$) and under strain ($\varepsilon = 50\%$) conditions. In order to clarify the breaking of H-bonds between in SMHIP polymer film, ionic liquid (IL) characteristics region ($1400\text{--}1000\text{ cm}^{-1}$, assigned to TFSI⁻ vibrational bands) and $3220\text{--}3050\text{ cm}^{-1}$ assigned to EMIM⁺ vibrational bands) and TPU characteristics region ($1760\text{--}1660\text{ cm}^{-1}$, assigned to TPU C=O stretching) are investigated. Supplementary Figure 10a shows the FT-IR characteristic regions of TFSI⁻ and EMIM⁺ of i(20)-silica-TPU SMHIP at normal state ($\varepsilon = 0\%$) and under external strain ($\varepsilon = 50\%$). It is important to note that most of the vibrational bands of TFSI⁻ (SO₂, CF₃, and S–N–S stretching) are shifted to higher frequencies under external strain. The shifting of TFSI⁻ vibrational bands towards higher frequencies is a reflection of bulk-like dynamics (*J. Phys. Chem. C* 2015, **119**, 24381; *Langmuir* 2013, **29**, 9744). Similarly, the vibrational bands of EMIM⁺ (C–H symmetric/asymmetric stretch, ring NC(H)NCH stretch, and CH₃(N)HCH stretch) are shifted to higher frequencies under strain. These results obtained here are in excellent agreements with the previous studies (*Polym. Bull.* **1991**, *25*, 491; *Mater. Today* **2014**, *17*, 57) dealing with spectroscopic investigations of polymeric materials under external deformations, given that the perturbations to the polymeric matrix caused by external deformations (pressure/strain) are reflected by the shifts of the IR bands. Hence, the spectroscopic results obtained for SMHIP films under external strain can be correlated to the weakening/breaking of TFSI/silica H-bonds and/or $\pi\text{--}\pi$ stacking interaction of imidazolium rings in SMHIP under external strain. Supplementary Figure 10b represents FT-IR spectra of C=O stretching region of TPU (where the two peaks at $\sim 1729\text{ cm}^{-1}$ and $\sim 1703\text{ cm}^{-1}$ correspond to the stretching vibration of free C=O and hydrogen-bonded C=O groups) of i(20)-silica-TPU SMHIP at normal state ($\varepsilon = 0\%$) and under strain ($\varepsilon = 50\%$). It is important to note that the intensity of hydrogen-bonded C=O groups is decreased significantly under strain while the intensity of free C=O is increased, which clearly reflects perturbations in SMHIP H-bond networks. We calculated H-bond index (ratio of the peak area of H-bonded -C=O groups/free -C=O groups of TPU) of i(20)-silica-TPU SMHIP film in the stretched state and compared with the upstretched condition. A pronounced decrease of $\sim 38.4\%$ in *cal.* H-bond index in the stretched state clearly signifies the breaking/weakening of H-bonds networks in SMHIP under external deformation. The spectroscopic investigation of SMHIP during strain clearly indicates the deformation of H-bond networks under external deformation. We reflected this on page 23 in revised supplementary information (**Figure S10**).

Supplementary Figure 10. FT-IR investigation of i(20)-silica-TPU under uniaxial stretching .
a ATR-FTIR spectra in the spectral regions 1400–1000 cm⁻¹ (corresponding to TFSI⁻ stretching) and 3200–3050 cm⁻¹ (corresponding to EMIM⁺ stretching); Results are shown for i(20)-silica-TPU SMHIP in normal state (no strain, $\varepsilon = 0\%$) (shown by black color) and during strain ($\varepsilon = 50\%$) condition (shown by red color). **b** ATR-FTIR spectra in the spectral region 1760–1660 cm⁻¹ (C=O stretching of TPU) in normal state (no strain, $\varepsilon = 0\%$) (shown by black color) and during strain ($\varepsilon = 50\%$) condition (shown by red color).

Q4) Some typo should be corrected.

Response: We thank the reviewer for pointing out typographical mistakes. We have corrected them.

Q5) Some other research articles (i.e., *J. Mater. Chem. B*, 2019,7, 173-197; *Adv. Funct. Mater.* 2019, 29, 1807343) should be cited.

Response: We appreciate the reviewer's comment. We have followed this suggestion and the suggested references are cited in the revised version (refs. no. 67 & 4).

[67]. Ma, Z., et al. *J. Mater. Chem. B*, 7, 173-197 (2019).

[4]. Li, S., et al. *Adv. Funct. Mater.* 29, 1807343 (2019).

Response to Reviewer #2

Thank you for your invaluable comments.

We revised the manuscript according to your comments.

Present manuscript entitled "A bioinspired, hydrogen bond-triggered ultrasensitive ionic mechanoreceptor skin" reports on the synthetic multicellular hybrid ion pump (SMHIP) film using ionic liquid (IL) 1-ethyl3-methylimidazoliumbistrifluorosulfonylimide [EMIM][TFSI] confined on silica microstructures embedded in thermoplastic polyurethane (TPU). The IL-silica-TPU SMHIP film has been developed by synthesizing IL-silica-gel through conventional sol-gel method followed by preparation of thermoplastic polyurethane (TPU) precursor gel and then preparing IL-silica-TPU solution heat treatment process. IL-silica-TPU film with 20 wt% IL resulted optimized film. 20 wt% is the optimized concentration to obtain the shape and size controlled IL-confined silica microstructures (called as artificial mechanoreceptor cells) Ionic skin sensor array was fabricated by sandwiching the IL-silica-TPU SMHIP film between a couple of patterned PDMS/Ag nanowire electrodes. The main pressure-sensing mechanism in the hybrid ionic skin was attributed to originated from the pumping of [EMIM][TFSI] ion pairs from surface of silica microstructures due to the pressure induced cleavage of H-bonds and /or pi-pi interactions in SMHIP similar to biological systems. The present research work may be helpful in synthesizing synthetic ionic mechanoreceptor skin for different applications. The manuscript can be accepted for publication subject to the following revisions.

Q1) *There should be more clarity in the introduction part.*

Response: We thank the reviewer for pointing this lack of clarity in the introduction. We added a few sentences to in the introduction part to impart clarity. We reflected this on page 2 & 3, in the revised manuscript. An ionic skin composed of deformable ionic materials represents a new class of deformable sensory platforms to emulate the tactile sensing features of human skin for potential applications in artificial skin technology (*Adv. Mater.* 2104, **26**, 7608; *Adv. Mater.* 2018, **30**, 1704403). Many ionic conductors such as ionic liquids (ILs), ionogels, and hydrogels, have been used to implement the ionic skin with human skin-like perceptive characteristics (*Lab Chip* 2014, **14**, 1107; *Macromol. Rapid Commun.* 2018, **39**, 1800246; *Nat. Rev. Mater.* 2108, **3**, 125). Therefore, the ionic skin can effectively sense pressure, strain, shear, torsion, and other external stimuli, but struggle to maintain high sensitivity over a wide spectrum of pressures in task specific applications such as robotics and prosthetics. Recently, biomimetics has emerged as a burgeoning area in artificial skin technology that has led innovations in material designing and device structure manipulation with the aim to imitate tactile sensing features of human skin intelligently (*Adv. Electron. Mater.* 2016, **2**, 1600356; *Nat. Mater.* 2012, **11**, 795). Biological cellular structures are the source of inspiration because of their intriguing structural and functional properties (*Adv. Mater.* 2017, **29**, 1605973; *ACS Nano* 2016, **10**, 4550).

[1]. Sun, J-Y., Keplinger C., Whitesides G. M., & Suo, Z. *Adv. Mater.* **26**, 7608–7614 (2014).

[2]. Lee, H-R., Kim, C-C., & Sun, J-Y. *Adv. Mater.* **30**, 1704403 (2018).

[3]. Nie, B., Li, R., Brandt, J. D., & Pan, T. *Lab Chip* **14**, 1107–1116 (2014).

- [4]. Li, S., et al. *Adv. Funct. Mater.* **29**, 1807343 (2019).
- [5]. Wang, H., Wang, Z., Yang, J., Xu, C., Zhang, Q., Peng, Z. *Macromol. Rapid Commun.* **39**, 1800246 (2018).
- [6]. Choi, D., et al. *Adv. Mater. Technol.* **4**, 1800284 (2018).
- [7]. Yang, C., & Suo. *Z. Nat. Rev. Mater.* **3**, 125–142 (2018).
- [8]. Pang, C. et al. *Nat. Mater.* **11**, 795–801 (2012).
- [9]. Ha, M. et al. *Adv. Funct. Mater.* **25**, 2841–2849 (2015).
- [10]. Kang, S. et al. *Adv. Electron. Mater.* **2**, 1600356 (2016).
- [11]. Chun, K-Y., Son, Y. J. & Han, C-S. *ACS Nano* **10**, 4550–4558 (2016).
- [12]. Jin, M. L. et al. *Adv. Mater.* **29**, 1605973 (2017).

Q2) Nyquist plots (Fig 5d) should have identical scales for Z' and Z'' axes.

Response: We appreciate the reviewer's comment. Nyquist plots shown in Fig. 5d have identical scales. Impedance Nyquist plots represent imaginary part $-Z''$ as a function of real part Z' . We reflected this on page 35, in the revised manuscript (**Figure 5d**).

Fig. 5d Impedance Nyquist plots (imaginary part $-Z''$ as a function of real part Z') of i(20)–TPU ionic polymer and i(20)–silica–TPU SMHIP piezocapacitive devices for no pressure (NP), under pressure (UP), and after removing pressure (AP).

Q3) Since the main pressure sensing mechanism in the proposed ionic skin has been attributed to the pumping of ionic liquid [EMIM][TFSI] ion pairs from surface of silica microstructures due to pressure induced cleavage of hydrogen bonds/ π - π interactions in SMHIP therefore, some references dealing with the interaction of cations/anions of ionic liquid molecules with the pore-wall surface of silica matrix are missing e.g. doi.org/10.1016/j.pmatsci.2014.03.001, [doi.10.1039/C4RA04362](https://doi.org/10.1039/C4RA04362), doi.org/10.1021/jp2003358.

Response: We thank the reviewer for pointing this out. Suggested references (doi.org/10.1016/j.pmatsci.2014.03.001 and doi.org/10.1021/jp2003358) are cited in the revised

version (refs. no. 31 & 32). Additional text related to these references is also included in the revised version. We believe that the reference (*doi.10.1039/C4RA04362*) is not relevant to our work. That's why we did not cite this reference in the revised manuscript.

[31]. Singh, M. P., Singh, R. K., Chandra, S. *Prog. Mater. Sci.* **64**, 73–120 (2014).

[32]. Singh, M. P., Singh, R. K., Chandra, S. *J. Phys. Chem. B* **115**, 7505–7514 (2011).

Response to Reviewer #3

We thank the reviewer for their careful reading of the manuscript and their constructive remarks. We revised the manuscript according to your comments.

The authors have reported “dynamic confinement” of charged molecules in an ionogel by introducing silica particles. Silica particles in the gel are making hydrogen bonding with bis(trifluoromethylsulfonyl)imide(TFSI-), so they could drop the conductivity of the ionic liquid. However, as the confinement of TFSI- could be released by a mechanical force, the gels could recover their conductivity when they undergo a mechanical deformation. Those kinds of dynamic capturing and releasing of ions by silica particles are firstly introduced. In that sense, this reviewer think the manuscript has enough novelty for the publication. Furthermore, because the authors have studied the confinements very well with EDX, Raman spectroscopy, and FT-IR, this reviewer think the manuscript is also well organized with sufficient verifications about their hypothesis. Therefore, this reviewer recommends the publication of this manuscript in Nature Communications. However, there are some unclear factors. I would like to ask and minor points which should be revised. Please consult the followings;

Q1) *In Fig. 2a, compared to carbon map, nitrogen map shows homogeneous distribution. Any ideas about the nitrogen map?*

Response: We thank the reviewer for asking us to clarify this important point. As mentioned in our work, we have developed a cellular-structured ionic polymer composite where the ionic liquid (IL) 1-ethyl-3-methylimidazoliumbis(trifluoromethyl-sulfonyl)imide ([EMIM]⁺[TFSI]⁻ cation–anion pairs) confined onto the surface of silica microstructures (dispersed phase) embedded in thermoplastic polyurethane (TPU) elastomeric matrix (continuous phase). It is important to note that nitrogen (N) is the constituent element of IL cation ([EMIM]⁺), IL anion ([TFSI]⁻), and even TPU (hard segments) as well. Further, due to embedded nature of IL-confined silica microstructures in the TPU polymer chains, these species (EMIM, TFSI, and TPU) co-exist in the silica region. Therefore, a uniform elemental distribution for N is observed in the silica region which signifies N signals coming from EMIM, TFSI, and TPU collectively. We reflected this on page 7, in the revised manuscript.

Q2) Similar with previous question. In Fig. 2d and 2e, the Raman intensity map of ‘ring-in-plane symmetric stretching’ is hard to understand. Any ideas about it?

Response: We appreciate the reviewer’s comment. The ‘ring-in-plane’ symmetric stretching refers to the imidazolium ring plane of 1-Ethyl-3-methylimidazolium ([EMIM]). Similar terminology is used frequently in various reports (*J. Phys. Chem. B* 2004, 108, 13177-13184; *J. Phys. Chem. A* 2014, 118, 6873-6882) dealing with electronic structures and spectroscopic investigations of imidazolium-based ionic liquids. In order to make clear understanding to the readers, these references are added in the revised manuscript (refs. no. 37 & 38).

[37]. Talaty, E. R., et al. *J. Phys. Chem. B*, **108**, 13177-13184 (2004).

[38]. Vyas, S., et al. *J. Phys. Chem. A*, **118**, 6873–6882 (2014).

Q3) EDX line scan profiles at Fig. 2b need y axis legend.

Response: We thank the reviewer for bringing up this important point. We have now added y-axis legends in Fig. 2b. We reflected this on page 31, in the revised manuscript (**Figure 2b**).

Fig. 2b EDX line scan profiles for C (K_{α}), Si (K_{α}), F (K_{α}), S (K_{α}), and N (K_{α}) along the noted path (green arrow, inset). X axis represents the distance along the scan line and Y-axis represents the relative elemental count along the scan line. Relatively higher intensity of F and S (coming from TFSI⁻) at circumferential region of silica structures and relatively high intensity of N (coming from TFSI, EMIM⁺, and TPU collectively) at the center of the silica microspheres observed in elemental maps and line scans establish a microscopic artificial plasma membrane structure consisting of ordered stepwise layers of TFSI⁻ and EMIM⁺ on silica microspheres (middle inset Fig. 2a, where grey sphere represents silica microstructures, blue anions represent TFSI⁻ ions and red cations represent EMIM⁺ ions).

REVIEWERS' COMMENTS:

Reviewer #1 (Remarks to the Author):

The reviewers addressed the questions well. Therefore, it is acceptable in current version.

Reviewer #2 (Remarks to the Author):

Suggested corrections have been incorporated in the revised manuscript and hence it can be accepted for publication. Scales are now identical between z prime and $-z$ double prime.

Reviewer #3 (Remarks to the Author):

All three minor points which I suggested are clearly solved by the author. This reviewer recommends the publication of this manuscript in Nature Communications.

Aug. 5th, 2019

REVIEWERS' COMMENTS:

Reviewer #1 (Remarks to the Author):

The reviewers addressed the questions well. Therefore, it is acceptable in current version.

Response: We thank the reviewer for his positive assessment to our work.

Reviewer #2 (Remarks to the Author):

Suggested corrections have been incorporated in the revised manuscript and hence it can be accepted for publication. Scales are now identical between z prime and $-z$ double prime.

Response: We thank the reviewer for his positive assessment to our work.

Reviewer #3 (Remarks to the Author):

All three minor points which I suggested are clearly solved by the author. This reviewer recommends the publication of this manuscript in Nature Communications.

Response: We thank the reviewer for his positive assessment to our work.